# School Parent Attitudes and Perceptions Relating to Animals, Animal-Assisted Interventions, and the Support of Children’s Mental Health

**DOI:** 10.3390/healthcare11070963

**Published:** 2023-03-28

**Authors:** Rhoda A. Leos, Paula M. Cuccaro, John R. Herbold, Belinda F. Hernandez

**Affiliations:** School of Public Health, University of Texas Health Science Center at Houston, Houston, TX 77030, USA; paula.m.cuccaro@uth.tmc.edu (P.M.C.);

**Keywords:** mental health, animal-assisted interventions, childhood trauma, school interventions

## Abstract

Growing awareness of the negative effects of trauma has led San Antonio, TX, school districts to expand efforts that can help mitigate these effects and support mental health. Given the literature around the psychological benefits of human–animal interactions, the concept of incorporating animals in treatments or interventions is not a new one. While schools have begun considering or utilizing animal-assisted interventions (AAIs), there have been limited efforts to understand existing perceptions relating to animals and AAIs among school parents in this Hispanic community. To address this gap, a cross-sectional study consisting of a 34-item survey was conducted to explore attitudes, knowledge, and perceptions relating to animals (i.e., pets), AAIs, and the need for supporting young children’s mental health among parents. A total of 187 surveys from two school districts were completed and utilized for analysis. The study’s findings demonstrate that parents acknowledged the importance of addressing mental health issues early on and were aware of the health benefits human–animal interactions can provide. Furthermore, parents had positive attitudes toward pets and positive perceptions toward AAIs in schools. Some implementation concerns were expressed relating to safety and well-being. Overall, these findings suggest there is existing parent support in using AAIs as a trauma-informed strategy and school innovation.

## 1. Introduction

San Antonio, Bexar County, is among many Texas communities that have experienced the burden of child abuse, violence, and trauma. In 2020, the rate of child abuse victims in Bexar County (10.3 per 1000) was higher than other counties, such as Dallas (9.8 per 1000), Harris (5.2 per 1000), and Travis (8.3 per 1000) [1]. In early 2021, a local news reporter highlighted police department reports of an 18 percent increase in calls related to family violence in 2020 [2]. Growing awareness of such data and the negative impact of trauma has driven the community to expand efforts that can help support and build resilience among children. This expansion has included the development of partnerships and collaboration between non-profits, the local health department, and school districts, as well participation in the South Texas Trauma-Informed Care Consortium to further understand the community’s needs.

Given the extensive literature on the positive psychological and physiological effects of human–animal interactions, it comes as no surprise that the use of animal-assisted interventions (AAIs) has also surfaced in the community as an innovation to support children’s mental health. AAIs can be defined as “goal-oriented and structured interventions that intentionally incorporate animals in health, education, and human service for the purpose of therapeutic gains and improved health and wellness” [3]. A 2017 systematic literature review highlighted the numerous positive benefits of having AAIs implemented in an educational setting. Such benefits include reductions in aggression, improved emotional stability, increased positive attitudes toward school, and enhanced learning [4]. Furthermore, AAI-related studies that have focused on children or adolescents with trauma have demonstrated positive mental health outcomes. One study found that treatment involving therapy dogs led to significant decreases in depression, anger, anxiety, and PTSD [5].

While San Antonio schools have begun considering or utilizing therapy dog visits and animal-assisted crisis response, there are no data to show existing perceptions and attitudes relating to animals and the use AAIs in San Antonio schools. When it comes to the process of adopting and implementing school-based interventions, parents are an important group of stakeholders who ultimately influence school district policies. They also play a crucial role when it comes to supporting children’s social–emotional development and mental health. Hence, learning and understanding existing perceptions, attitudes, and knowledge that currently exist among this key group is an important step. As an effort to address the gap in research, the current study explored attitudes, knowledge, and perceptions related to animals/pets, AAIs, and the need for supporting young children’s mental health among parents in the San Antonio community.

## 2. Materials and Methods

### 2.1. Study Design/Approach

A cross-sectional exploratory research design was utilized to investigate attitudes, knowledge, and perceptions among parents of children of school-aged children (including pre-kindergarten and kindergarten) in San Antonio. Specific constructs within these areas were assessed through a short survey consisting of Likert-scale questions and one open-ended question.

### 2.2. Survey and Measures

The first half of the survey assessed parents’ attitudes toward pets (pet attitudes) utilizing a modified format of the Pet Attitude Scale (PAS), a self-report tool consisting of 18 question items in a Likert format. The PAS is one of the first published scales to assess attitudes toward companion animals and one of the few scales with reliability information (Cronbach’s alpha of 0.93 and test–retest reliability of 0.92) [6]. Additional research has supported the construct validity of the scale and has showcased it as an example of a good and suitable psychometric instrument [7] (p. 138). The question items in this scale include, “You should treat your house pets with as much respect as you would a human member of your family” and “Having pets is a waste of money” [8] (pp. 351–352). In the modified format of the scale the words “or would if I had one” were added to items 2, 8, and 16 to apply to individuals who do not have a pet. Response options ranged from 1 (strongly disagree) to 7 (strongly agree) and each item is scored with the corresponding number. Individual responses are then summed to obtain a total score for pet attitude (ranging from 18 to 126), with items 4, 6, 9, 12, 13, and 17 reversed scored (i.e., a response of 7 is scored as 1). The higher the score, the more positive the individual’s feelings/attitudes toward companion animals [6].

The second half of the survey consisted of question items developed by the PI to assess four additional measures. Items 19 and 21 assessed the perceived importance of addressing mental/emotional health issues among young children (mental health perceptions; Cronbach’s alpha of 0.71). Example items include, “It’s important to address mental or emotional health issues, including the effects of trauma, among young children”. Items 23–27 assessed existing knowledge of AAIs and the positive effect of animals (AAI knowledge; Cronbach’s alpha of 0.80). Example items include, “Animals/pets can help strengthen social skills, such as empathy and relationship-building” and “I know what animal-assisted interventions are and how they are utilized”. Questions 28, 30–32, and 34 assessed perceptions of AAIs in schools (AAI perceptions; Cronbach’s alpha of 0.71). Example items include “Animal-assisted interventions would be more beneficial to students than other school programs/activities that are currently being implemented in my child’s school” and “Having animal-assisted interventions in my child’s school would concern me”. These questions were developed in a Likert format to follow the same scoring method as the PAS items. Hence, higher scores indicate greater perceived importance of addressing mental health issues among young children, higher knowledge related to the positive effect of animals and the definition of AAIs, as well as more positive perceptions relating to the use of AAIs in schools.

Supplementary questions were included for additional information relating to AAI perceptions and mental health perceptions. The questions included: “How early [which grade level] should these [mental/emotional health] issues begin to be addressed?”, “I am aware of programs/services that are currently offered within my child’s school that help support mental or emotional health”, “In which grade level would animal-assisted interventions be most beneficial?”, and “If you have concerns with having animal-assisted interventions in your child’s school, what would be your main concern?”. These questions were developed as multiple-choice questions, apart from one open-ended question. See Appendix A for the finalized survey.

### 2.3. Recruitment and Data Collection

Given the existing AAI literature supporting positive outcomes among young school children, parents of preschool and elementary school children within San Antonio school campuses were recruited. Due to the previously established relationships and agreements with specific campuses in two school districts, a convenience sample of parents was recruited from these campuses consisting of predominantly Hispanic/Latino students from economically disadvantaged households. From the eight participating campuses, two were early childhood education centers with grade levels up to second grade. To have consistent parameters across campuses, parents with one or more children enrolled in PreK through 2nd grade were eligible to participate.

Recruitment and data collection took place during the Spring 2021 school semester. Given the impact of the COVID-19 pandemic and restricted access to school campuses, parents were contacted via email as a primary method for recruitment and were able to complete the survey online (via Qualtrics online survey software). Recruitment emails were sent by school administration with a brief description of the study and survey link. The survey link directed participants to a letter of information and a screening question to confirm eligibility. Once a participant reviewed the letter of information and confirmed eligibility (checking off that they were a parent of one or more children enrolled in a participating campus), they could proceed to complete the demographic questions and survey. Demographic data requested included gender, age, ethnicity, race, education level, number of children enrolled, and number of pets/animals in the household.

While most students participated in virtual learning during the 2020–2021 school year, a small percentage of students continued in-person learning at their campus. As an additional method of recruitment, paper surveys were sent home with this specific group of students. Included with the paper survey were the letter of information and a blank envelope for parents to return their completed surveys. At least one reminder was sent to parents via email and those who completed 50% or more of the survey and provided their contact information (either online or on paper) were entered into a drawing for a USD 50 gift card. The study’s protocol was reviewed and approved by the UTHealth School of Public Health’s Committee for the Protection of Human Subjects (CPHS), as well as school district administrators and their research committee.

### 2.4. Data Analysis

The paper and online survey responses were combined and entered into an Excel spreadsheet and reviewed for errors. Demographic errors, such as invalid age, were coded as missing. Total scores for pet attitude, mental health perceptions, AAI knowledge, and AAI perceptions were calculated and entered for each participant. Participants who did not complete all questions for a specific measure had their total score coded as missing. Upon cleaning and finalizing the data, descriptive statistics were utilized to summarize participant demographics (overall and by school district). Statistics included counts and percentages to summarize information on gender, ethnicity, race, education level, number of children enrolled, and pet ownership, as well as means to summarize age. Means were used to summarize scores for pet attitudes, mental health perceptions, AAI knowledge, and AAI perceptions. Counts and percentages were also used to summarize responses for supplementary questions.

Chi-squared tests were conducted to assess whether there were significant differences in demographic characteristics by school district and whether there were significant differences in responses for supplementary questions by gender, ethnicity, and pet ownership. Data assumptions for a two-sample t-test were assessed in order to further investigate whether there were any significant differences in (1) scores for pet attitudes by gender, ethnicity, and pet ownership; (2) scores for mental health perceptions by gender, ethnicity, and pet ownership; (3) scores for AAI knowledge by gender, ethnicity, and pet ownership; and (4) scores for AAI perceptions by gender, ethnicity, and pet ownership. Since assumptions of normality were violated, a nonparametric test was utilized in place of the two-sample t-test. Pearson correlation coefficients were also calculated to assess the relationship between scores for pet attitudes, AAI perceptions, and AAI knowledge.

Finally, a thematic analysis was conducted by the PI to analyze responses for the open-ended question: “If you have concerns with having animal-assisted interventions in your child’s school, what would be your main concern?” For this analysis, all paper and online responses were initially combined and entered in an MS Word document. The following steps of the analysis process involved reading and re-reading all participant responses and organizing the data into meaningful chunks, which were assigned descriptive codes. The resulting codes were then grouped together to form key themes. In the final steps of the analysis, the PI reviewed the themes to ensure they were clear and distinct from one another. Specific quotes to help represent and describe each theme were also identified for reporting.

## 3. Results

### 3.1. Participant Demographics

A total of 200 parents across both districts completed or initiated the survey (55 on paper and 145 through the survey link). Given the lag in student enrollment data provided by the Texas Education Agency (the governing body overseeing primary and secondary public education in the state), the total numbers for students enrolled in PreK–2nd grade, as well as the total numbers of parents across each school district, were not made available. Hence, the total number of parents that were non-respondent is not reported. Those who completed less than 50% of the survey were excluded, leaving a total sample size of 187 parents. As shown in Table 1, the average age of parents was 35.56 years with the majority identifying as female (78.61%), Hispanic or Latino (77.35%), and white (64.64%). The majority also had a high school diploma or GED as their highest level of education (71.74%). In addition, most parents had one child enrolled in either district (64.52%) and had an animal or pet at home (65.24%). The results for chi-square tests indicated there were significant differences in race, ethnicity, education, and pet ownership between districts A and B. The results for the Wilcoxon rank-sum test indicated that there were no significant differences in age between school districts.

### 3.2. Survey Scores

Mean scores for pet attitudes, mental health perceptions, AAI knowledge, and AAI perceptions are summarized in Table 2. The participants that did not respond to all questions were not able to be scored for one or more variables (missing scores resulted in different counts for each variable). The overall average score for pet attitudes was 101.62 (SD = 15.36) out of a maximum possible score of 126. The overall average score for mental health perceptions was 13.34 (SD = 1.45) out of a maximum possible score of 14. The overall average score for AAI knowledge was 29.53 (SD = 4.63) out of a maximum possible score of 35. The average score for AAI perceptions was 27.89 (SD = 4.72) out of a maximum possible score of 35.

There were no significant differences in scores for pet attitudes, mental health perceptions, AAI knowledge, and AAI perceptions when comparing males vs. females and Hispanics vs. non-Hispanics. When it came to pet ownership, those who owned a pet had higher scores for pet attitude (mean = 105.53, SD = 12.80), AAI knowledge (mean = 30.43, SD = 4.29), and AAI perceptions (mean = 28.79, SD = 4.85) compared to those who did not own a pet. These differences were all statistically significant (*p* < 0.001). While those who owned a pet also had higher scores for mental health perceptions (mean = 13.5, SD = 1.03) compared to those who did not (mean = 13.03, SD = 1.97), this difference was not statistically significant (*p* > 0.05)

### 3.3. Supplementary Questions Relating to Mental Health Perceptions

Almost 66% of participants indicated that mental or emotional health issues, including the effects of trauma, should begin to be addressed in pre-kindergarten. Approximately 50.27% of parents indicated some level of agreement (either slight, moderate, or strong) in terms of their awareness of current school-based programs or services that support mental or emotional health. The chi-square test results demonstrate that there were no statistically significant differences in the responses by gender, ethnicity, and pet ownership.

### 3.4. Supplementary Questions Relating to AAI Perceptions

Almost 66% of parents indicated that AAIs would be most beneficial across all grade levels versus a specific grade level. The chi-square test results demonstrate that there were no statistically significant differences in responses by ethnicity and pet ownership. However, when comparing males and females, more females indicated that AAIs would be beneficial across all grade levels. As shown in Table 3, this comparison resulted in a statistically significant difference (*p* = 0.012).

### 3.5. Correlation between Pet Attitudes, AAI Perception, and AAI Knowledge

The correlation coefficients provided evidence for a moderate positive correlation between scores for pet attitudes and AAI perceptions (r = 0.47, *p* < 0.001), scores for pet attitudes and AAI knowledge (r = 0.48, *p* < 0.001), as well as scores for AAI knowledge and AAI perceptions (r = 0.53, *p* < 0.001). Particularly, more positive attitudes toward pets was related to more positive perceptions toward AAIs and more knowledge of AAIs and the benefits of animals. In addition, more knowledge of AAIs was related to more positive perceptions toward AAIs. These results are provided in Table 4.

### 3.6. Concerns about AAIs in Schools

While not all parents chose to share concerns about having AAIs in school, there were three key themes found among those that did. These themes included: children’s safety and well-being, impact on academics, and animal welfare. In relation to children’s safety and well-being, most responses referenced the fear of a child being bitten. Words such as “snapping” were used to convey the potential reaction of a dog. While allergies also came up as a factor that could compromise well-being for some students, the fear of being physically harmed seemed to be more prominent in this theme. When it came to the concerns relating to academics, participants acknowledged that having an animal on campus would be something “out of the norm” that could easily distract students. One parent stated, “…the animal would cause a distraction for kids to keep focus on their studies. They would probably be playing and trying to pet the animal”. Concern about animals’ well-being during the implementation of AAIs was made evident, as one parent commented, “my biggest concern would be that the children were monitored at all times…to ensure that none of the animals are ever treated cruelly or harmed or frightened in any way by the children”. These three resulting themes could not be further dissected given the limited data provided in participant responses.

## 4. Discussion

The value of supporting mental and emotional health within the San Antonio school community was made evident as we found that most participating parents strongly agreed that addressing mental or emotional health issues (including the effects of trauma) and having access to school-based programs are important. The importance and value that is placed on addressing mental health serves as a segue for introducing novel interventions that can address trauma and support children’s mental health. Findings further demonstrated that parents overall had positive attitudes toward pets, existing knowledge of AAIs and the positive effect of animals, as well as positive perceptions of AAIs in schools. Pet owners had the most positive perceptions and attitudes toward pets and AAIs, and the belief that AAIs would be beneficial across all grade levels was more prominent among females. Furthermore, it was found that those with more knowledge of AAIs and more positive pet attitudes also held more positive perceptions of AAIs in schools.

This study is the first to assess knowledge, perceptions, and attitudes relating to animals and AAIs in schools among a predominantly Hispanic population. Hence, it is also the first AAI-related study to consider native Spanish speakers with a translated version of the Pet Attitude Scale. Finding that parents in this population perceive animals and AAIs in a positive way helps affirm the potential of introducing AAIs to support minority children. Considering the disparities that exist in mental health service utilization, implementing AAIs in an educational setting may be an avenue to reach those children who may not be receiving mental health support outside a school setting or that may be less likely to seek treatment later in life. Furthermore, AAIs in schools could help reduce some mental health conditions and the need for mental health services in these populations. These effects are important to consider given the shortage of mental health services available for children [9].

Our findings demonstrating positive perceptions toward AAIs further align with previous studies that have evaluated specific school-based AAIs and have found positive perceptions among parents and key stakeholders (i.e., counselors and teachers) [10,11,12,13,14,15]. Pet owners in our sample population had more positive attitudes toward animals, which supports early literature around closeness to animals and how pet ownership is important in shaping attitudes toward animals [16]. Those with positive pet attitudes in our sample had more positive perceptions toward AAIs, which supports previous research where those with positive attitudes toward pets or companion animals found AAIs to be more positive and credible than those with negative pet attitudes [17]. We found no gender differences when it came to pet attitudes, which may not seem to align with previous studies that have compared males vs. female attitudes toward companion animals. However, Herzog [18] makes the point that males and females are more similar than different when it comes to human–animal interactions (e.g., their desire to live with an animal). Furthermore, our findings relating to concerns around AAIs in schools are consistent with previous research relating to challenges or barriers in implementing AAIs. Issues such as allergies, fear of animals, and animal welfare have surfaced in previous studies [14,19,20].

Considering the use of convenience sampling and a limited sample size, we cannot generalize our findings to all parents or school districts. We recognize the presence of bias in our sample, consisting primarily of females and pet owning parents. Research with a larger and more diverse sample is needed to further confirm existing positive attitudes and perceptions in the community and further explore gender differences in AAI perceptions. We further recognize that the term “animal-assisted interventions” was used as a broad term and did not consider if or how perceptions may vary depending on the type of AAI or type of animal that is used (e.g., a therapy dog visit vs. a dog reading program, the use of a rabbit vs. a dog). Different AAIs may be a better fit for some schools or school districts than others, and future studies around specific AAIs may be of greater value to some groups.

## 5. Conclusions

Our findings reveal that most parents, including ethnic minorities, may already have some knowledge of the mental and social–emotional benefits that animals can offer and would support the implementation of AAIs in schools. While AAIs have been used to help trauma-exposed children, the benefits of human–animal interactions extend to the larger population of school-aged children. Thus, this research not only highlights AAIs as an intervention opportunity to mitigate the effects of early trauma, but an intervention that can accompany existing efforts that protect school children’s mental health and social emotional development. Furthermore, we emphasize the value in engaging parents and other key stakeholders in the implementation of such interventions and educating groups that may have limited knowledge of AAIs or less positive attitudes toward animals. Future evaluative studies will be instrumental in education efforts and in identifying best practices for AAIs in school settings.

## Figures and Tables

**Table 1 healthcare-11-00963-t001:** Summary of the participant demographics—overall and by school district.

		n (%)			*p*-Value
		District A	District B	Total	
Gender	Male	23 (21.30)	17 (21.52)	40	0.971
	Female	85 (78.70)	62 (78.48)	147	
Ethnicity	Hispanic/Latino	94 (87.04)	46 (58.23)	140	* <0.001
	Not Hispanic/Latino	14 (12.96)	27 (34.18)	41	
	No response		6 (7.59)		
Race	White	81 (75.00)	36 (45.57)	117	* <0.001
	Black or African American	7 (6.48)	27 (34.18)	34	
	Unknown/unsure	6 (5.56)	3 (3.80)	9	
	Prefer not to answer	14 (12.96)	7 (8.86)	21	
	No response		6 (7.59)		
Education	Some high school	5 (4.63)	13 (16.46)	18	* 0.009
	High school diploma	77 (71.30)	55 (69.62)	132	
	Bachelor’s degree	21 (19.44)	7 (8.86)	28	
	Master’s degree	5 (4.63)	1 (1.27)	6	
	No response		3 (3.80)		
Children enrolled in participating school	1	64 (59.26)	56 (70.89)	120	0.045
	2	26 (24.07)	18 (22.78)	44	
	3 or more	18 (16.67)	4 (5.06)	22	
	No response		1 (1.27)		
Pets	Yes	82 (75.93)	40 (50.63)	122	* <0.001
	No	26 (24.07)	39 (49.37)	65	
Age mean (SD)		36.56 (12.18)	34.02 (8.35)	35.52 (10.82)	0.655

* Statistically significant difference between groups (at the 0.05 level).

**Table 2 healthcare-11-00963-t002:** Mean scores for pet attitudes, mental health perceptions, AAI knowledge, and AAI perceptions—overall, by gender, ethnicity, and pet ownership.

		Pet Attitudes (n = 170)		Mental Health Perceptions (n = 178)		AAI Knowledge (n = 180)		AAI Perceptions (n = 175)	
		Mean (SD)	*p*-Value	Mean (SD)	*p*-Value	Mean (SD)	*p*-Value	Mean (SD)	*p*-Value
Overall Summary		101.62 (15.36)		13.34 (1.45)		29.53 (4.63)		27.87 (4.72)	
Gender	Male	100.25 (15.97)	0.457	13.27 (1.02)	0.071	29.35 (4.46)	0.760	27.23 (4.33)	0.248
	Female	101.99 (15.24)		13.35 (1.54)		29.57 (4.67)		28.03 (4.81)	
Ethnicity	Hispanic/Latino	102.69 (14.74)	0.350	13.36 (1.32)	0.884	29.69 (4.16)	0.565	28.24 (4.28)	0.561
	Not Hispanic/Latino	99.66 (16.45)		13.62 (0.68)		29.64 (5.77)		27.15 (5.91)	
Pets	Yes	105.53 (12.80)	<0.001 *	13.5 (1.03)	0.091	30.43 (4.29)	<0.001 *	28.79 (4.85)	<0.001*
	No	94.81 (17.10)		13.03 (1.97)		27.86 (4.80)		26.24 (4.01)	

* Statistically significant difference between groups (at the 0.05 level).

**Table 3 healthcare-11-00963-t003:** School grade perceived to be ideal for AAIs—overall, by gender, ethnicity, and pet ownership.

“In Which Grade Level Would Animal-Assisted Interventions Be Most Beneficial?”		n (%)							*p*-Value
		PreK	Kindergarten	1st	2nd	After 2nd	All Grade Levels	Multiple Grades (PreK–2nd)	
Overall summary (n = 173)		24 (13.87)	9 (5.20)	12 (6.94)	1 (.58)	2 (1.16)	114 (65.90)	11 (6.36)	
Gender	Male (n = 36)	3 (8.33)	4 (11.11)	2 (5.56)	1 (2.78)	2 (5.56)	21 (58.33)	3 (8.33)	0.012 *
	Female (n = 137)	21 (15.33)	5 (3.65)	10 (7.30)	0 (0.00)	0 (0.00)	93 (67.88)	8 (5.84)	
Ethnicity	Hispanic/ Latino (n = 128)	17 (13.28)	8 (6.25)	7 (5.47)	1 (0.78)	2 (1.56)	85 (66.41)	8 (6.25)	0.898
	Not Hispanic/ Latino (n = 39)	7 (17.95)	1 (2.56)	2 (5.13)	0 (0.00)	0 (0.00)	26 (66.67)	3 (7.69)	
Pets	Yes (n = 113)	11 (9.73)	5 (4.42)	8 (7.08)	1 (0.88)	2 (1.77)	78 (69.03)	8 (7.08)	0.344
	No (n = 60)	13 (21.67)	4 (6.67)	4 (6.67)	0 (0.00)	0 (0.00)	36 (60.00)	3 (5.00)	

* Statistically significant difference between groups (at the 0.05 level)

**Table 4 healthcare-11-00963-t004:** Correlation between AAI perceptions, pet Attitudes, and AAI knowledge.

	Pet Attitudes	AAI Perceptions	AAI Knowledge
Pet attitudes	1.0		
AAI perceptions	0.4733 **	1.0	
AAI knowledge	0.4829 **	0.5323 **	1.0

** *p* < 0.001.

## Data Availability

The data presented in this study are available upon request from the corresponding author. The data are not publicly available due to the fact of privacy.

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
