# Peer review of "School Parent Attitudes and Perceptions Relating to Animals, Animal-Assisted Interventions, and the Support of Children’s Mental Health"

_healthcare, 2023, doi:10.3390/healthcare11070963_

Round 1
Reviewer 1 Report
This research is focused on examining parents` attitudes, perceptions and knowledge related to animals and animal-assisted interventions and children`s mental health.
The topic is relevant, especially since the need for support and help to school aged children in their local community is recognized (the authors show the prevalence of violence in their area).
The manuspcript presents with various strenghts which include most notably the variables under examination. Despite these strengths it suffers from a few problems, which if adressed could strengthen the manuscript.
The variable „mental health of children“ is very superficial. It seems more as an assessment of aprental perception about the need for protecting mental health of children.
The subsection Data collection (p.3) is too broad. More precisely, the first paragraph (line 114-126) gives too much general data.
In the subsection Results, there are too many large tables, and the data within them is repeated in the text that follows.
Again, table 3a and 3b are unnecessary because the items are not directly related to mental health in children
The Discussion subsection could be improved in the sense that the autors should specifically compare their single results with previous research findings.
It should be highlighted that the sample is biased, because the sample consists of almost twice as many participants who has pets, so we can assume that their attitudes would be more positive to animal in general. It would be useful to shortly comment on the importance and benefits of animals for children who were not traumatized, or rather their usefulness for school aged children in general.
Author Response
The revised manuscript has been included as an attachment.
Point 1: The variable “mental health of children“ is very superficial. It seems more as an assessment of parental perception about the need for protecting mental health of children.
We recognize this construct was described vaguely in our abstract and at the beginning of our paper. We have revised the wording to better represent what we are assessing (see updated lines below):
Line 15-16 (abstract): “…to explore attitudes, knowledge, and perceptions relating to animals (pets), AAIs, and the need for supporting young children’s mental health among parents.”
Lines 62-65: “As an effort to address the gap in research, the current study explored attitudes, knowledge, and perceptions related to animals/pets, AAIs, and the need for supporting young children’s mental health among parents in the San Antonio community.”
We also updated our title to read: School Parent Attitudes and Perceptions relating to Animals, Animal-Assisted Interventions, and the Support of Children’s Mental Health
We kept the variable name/label “mental health perceptions” (underlined section 2.2 and utilized in the rest of the manuscript) for conciseness.
Point 2: The subsection Data collection (p.3) is too broad. More precisely, the first paragraph (line 114-126) gives too much general data.
We have removed the broader data from the first paragraph in section 2.3 and included additional content specific to our sample and inclusion criteria.
Point 3: In the subsection Results, there are too many large tables, and the data within them is repeated in the text that follows.
We have removed tables 3a and 3b. Table 3c was re-labeled as Table 3.
Point 4: Again, table 3a and 3b are unnecessary because the items are not directly related to mental health in children.
We have removed tables 3a and 3b. Table 3c was re-labeled as Table 3.
Point 5: The Discussion subsection could be improved in the sense that the authors should specifically compare their single results with previous research findings.
We have revised the third paragraph in the discussion section (starting on line 742) to better compare specific results to previous research findings.
Point 6: It should be highlighted that the sample is biased, because the sample consists of almost twice as many participants who have pets, so we can assume that their attitudes would be more positive to animals in general.
We have updated the last paragraph in the discussion section to highlight this limitation.
Point 7: It would be useful to shortly comment on the importance and benefits of animals for children who were not traumatized, or rather their usefulness for school aged children in general.
We appreciate this suggestion and have updated our conclusion to make mention of the usefulness of AAIs in the larger population of school age children.

Reviewer 2 Report
Authors identified an important community concern as a rationale for conducting this study. While it has most relevance for a city with high child abuse, such as San Antonio, it is useful for a broad group of cities less plagued by child abuse. However, addressing an outcome of child abuse seems less valuable than addressing the child abuse itself. It might strengthen the manuscript to acknowledge this as a secondary type of intervention that might accompany other efforts to support parent mental health and self-regulation. Animal-assisted interventions have broader benefits that could support children who have not experienced child abuse, suggesting a broader value for conducting this study. Addition of this type of information might increase the value of this manuscript.
Abstract line 18 & 19: Change “towards” to “toward.”
Pg 2, line 54: change “that” to “who.”
Pg. 2+, line 69, 72, 84, 299, 301, 323, 325, 327: Change “towards” to “toward.”
Pg 4, line 183, Change “lastly” to “finally.”
Pg 4, line 197, change “that” to “who.”
Pg. 5, Table 1: percentages do not seem correct. To be more useful, percentages should be for each District, not across both districts. (e.g.: Male from District A should be 23 (21.30))
Pg. 12, Survey title: Change “towards” to “toward.”
Author Response
The revised manuscript has been included as an attachment.
Point 1: Authors identified an important community concern as a rationale for conducting this study. While it has most relevance for a city with high child abuse, such as San Antonio, it is useful for a broad group of cities less plagued by child abuse. However, addressing an outcome of child abuse seems less valuable than addressing the child abuse itself. It might strengthen the manuscript to acknowledge this as a secondary type of intervention that might accompany other efforts to support parent mental health and self-regulation. Animal-assisted interventions have broader benefits that could support children who have not experienced child abuse, suggesting a broader value for conducting this study. Addition of this type of information might increase the value of this manuscript.
We appreciate this feedback and have made some updates to our conclusion that directly mentions the broader value of AAIs and human-animal interactions.
Point 2: Abstract line 18 & 19: Change “towards” to “toward.”
Change has been made.
Point 3: Pg 2, line 54: change “that” to “who.”
Change has been made.
Point 4: Pg. 2+, line 69, 72, 84, 299, 301, 323, 325, 327: Change “towards” to “toward.”
Change has been made.
Point 5: Pg 4, line 183, Change “lastly” to “finally.”
Change has been made.
Point 6: Pg 4, line 197, change “that” to “who.”
Change has been made.
Point 7: Pg. 5, Table 1: percentages do not seem correct. To be more useful, percentages should be for each District, not across both districts. (e.g.: Male from District A should be 23 (21.30))
All percentages in Table 1 have been updated.
Point 8: Pg. 12, Survey title: Change “towards” to “toward.”
Change has been made.

Reviewer 3 Report
This study address a potentially interesting topic, however I have several concernes. I report my comments:
-the aim of the study is not well reported and study's hypothesis were not reported
-The enrollement procedure is not clear. The authors reported about 1800 students and less than 200 participants...this is a very selected sample that represents a bias that may strongly affect the importance and generability of the findings. Moreover I suggest to clearly report the rate regarding not respondant, drop out, not eligible case etc.
-table 2: a) in column the authors reported different number of participants whereas the authors reported 187 participant but in the column are less than 187 for every variable; b) as the authors reported their finding in table the only significant difference is between Pets yes/no in the Pet attitude, AAI Knowledge and attitude whereas in the text the authors reported a series of differences not statistically significant (as authors declare) and this is to avoide since is highly confusing.
-Table 4 I have some concerns. Linear regression is usually performed to assess the predictive effect of a series of independent variable on the dependent one. As reported in the data analysis section by the authors it seems that the aim was more coherent with a correlation analysis than a regression one. Moreovere in the table the authors did not report standardized coefficient (beta) that are important and should be provided.
-Thematic analysis of the open questions are difficult to understand I suggest to report it more clerarly and to add some examples.
Author Response
The revised manuscript has been included as an attachment.
Point 1: the aim of the study is not well reported and study's hypothesis were not reported
This was exploratory research and our aim was to explore parent attitudes, knowledge, and perceptions relating to animals (pets), animal-assisted interventions, and the need for supporting young children’s mental health (this statement has been provided in lines 14-16 of the abstract and in lines 62-65 of the introduction).
Point 2: The enrollment procedure is not clear. The authors reported about 1800 students and less than 200 participants...this is a very selected sample that represents a bias that may strongly affect the importance and generability of the findings. Moreover, I suggest to clearly report the rate regarding non respondent, drop out, not eligible cases, etc.
We originally mentioned estimates of enrolled students at each school district, however, these numbers were based on data from the previous year provided by the Texas Education Agency (the governing body overseeing primary and secondary public education in our state). We also did not have access to data that would indicate whether each student had two parent/guardians listed as contacts or just one (administrators sent the recruitment emails). Hence, we were not able to report the total number of parents in our population nor the total number of parents that were non-respondent. We have updated our results (first paragraph) to include this information and provide more clarity.
Point 3: Table 2: a) in column the authors reported different number of participants whereas the authors reported 187 participants but in the column are less than 187 for every variable.
The different counts (n values) are due to those participants that did not respond to all the questions in each subsection. With missing responses, these participants were not able be scored appropriately for one or more variables. We have updated our results (first paragraph) to include this information.
Point 4: Table 2: b) as the authors reported their finding in table the only significant difference is between Pets yes/no in the Pet attitude, AAI Knowledge and attitude whereas in the text the authors reported a series of differences not statistically significant (as authors declare) and this is to avoid since is highly confusing.
The second paragraph of subsection 3.2 was removed to avoid confusion and lines 468-470 were added.
Point 5: Table 4 I have some concerns. Linear regression is usually performed to assess the predictive effect of a series of independent variable on the dependent one. As reported in the data analysis section by the authors it seems that the aim was more coherent with a correlation analysis than a regression one. Moreover, in the table the authors did not report standardized coefficient (beta) that are important and should be provided.
We have replaced our linear regression with a correlation analysis. This change is reflected in the following sections of the manuscript:
- Second paragraph of section 2.4 (line 223-224)
- First paragraph of section of 3.5 (starting on line 534)
- Table 4. Correlation between AAI Perceptions, Pet Attitudes, and AAI Knowledge (starting line 534)
Point 6: Thematic analysis of the open questions are difficult to understand I suggest reporting it more clearly and to add some examples.
The thematic analysis yielded limited findings given the short number of comments that were provided by parents and the conciseness of those comments (only 1 or 2 short sentences). We have updated the paragraph under section 3.6 within the results to make the content clearer. The quotes from parents are included to provide examples that support each theme.

Round 2
Reviewer 3 Report
I have carefully read the manuscript and I found it improved and acceptable for publication. The modified version is easier to read and cleared than the previous one.